

# CURTAINs flows for flows: Constructing unobserved regions with maximum likelihood estimation

Debajyoti Sengupta⋆, Sam Klein†, John Andrew Raine‡ and Tobias Golling°

Université de Genève, Genève, CH-1211 Switzerland

⋆ debajyoti.sengupta@unige.ch , † samuel.klein@unige.ch ,
‡ john.raine@unige.ch , ° tobias.golling@unige.ch

## Abstract

Model independent techniques for constructing background data templates using generative models have shown great promise for use in searches for new physics processes at the LHC. We introduce CURTAINsF4F, a major improvement to the CURTAINs method by training the conditional normalizing flow between two side-band regions using maximum likelihood estimation instead of an optimal transport loss. The new training objective improves the robustness and fidelity of the transformed data and is much faster and easier to train. We compare the performance against the previous approach and the current state of the art using the LHC Olympics anomaly detection dataset, where we see a significant improvement in sensitivity over the original CURTAINs method. Furthermore, CURTAINsF4F requires substantially less computational resources to cover a large number of signal regions than other fully data driven approaches. When using an efficient configuration, an order of magnitude more models can be trained in the same time required for ten signal regions, without a significant drop in performance.



# 1   Introduction

The search for new physics phenomena is one of the cornerstones of the physics programme at the Large Hadron Collider (LHC). The unparalleled energy and intensity frontier provided by the LHC provides a huge range of phase space where new signatures may be observed. The ATLAS [1] and CMS [2] collaborations at the LHC perform a wide array of searches for new particles beyond the standard model (BSM) of particle physics. Many of these searches target specific models, however, due to the vast possibilities of models and particles, dedicated searches cannot be performed for all possible scenarios.

Model independent searches aim to provide a broad sensitivity to a wide range of potential BSM scenarios without targetting specific processes. A key technique used in many searches is the bump hunt. Under the assumption that a new BSM particle is localised to a certain mass value, a bump hunt scans over an invariant mass distribution looking for excesses on top of a smooth background. Bump hunts were crucial in the observation of the Higgs boson by the ATLAS and CMS collaborations [3,4]. However, despite the success at finding the Higgs boson, there is little evidence for any BSM particles at either experiment [5–10]. With advances in machine learning (ML) many new model independent methods have been proposed to enhance the sensitivity to BSM physics [11–29] including approaches which aim to improve the sensitivity of the bump hunt itself [30–37].

In this work we improve upon the CURTAINs approach [35] by replacing the optimal transport loss used to train a flow between two complex distributions with maximum likelihood estimation. In order to evaluate the likelihood of the complex distributions on either side of the normalizing flow, we use the *Flows for Flows* technique introduced in Ref. [38] and applied to physics processes in Ref. [39]. This new configuration is called CURTAINsF4F.

We apply CURTAINsF4F to the LHC Olympics (LHCO) R&D dataset [40], a community challenge dataset for developing and comparing anomaly detection techniques in high energy physics [23]. We compare it to the previous iteration of CURTAINs, as well as to a current state of the art data driven approach CATHODE [32]. We evaluate the performance both in terms of improved signal sensitivity, but also in the required computational time to train the background models for a number of signal regions.

# 2   Dataset

We evaluate the performance of CURTAINsF4F using the LHC Olympics R&D dataset.

The LHCO R&D dataset [40] comprises background data produced through QCD dijet production, with signal events arising from the all-hadronic decay of a massive particle to two other massive particles $W' \rightarrow X(\rightarrow q\bar{q})Y(\rightarrow q\bar{q})$, each with masses $m_{W'} = 3.5$ TeV, $m_X = 500$ GeV, and $m_Y = 100$ GeV. Both processes are simulated with Pythia 8.219 [41] and

interfaced to `Delphes` 3.4.1 [42] for detector simulation. Jets are reconstructed using the anti-$k_\mathrm{T}$ clustering algorithm [43] with a radius parameter $R = 1.0$, using the `FastJet` [44] package. In total there are 1 million QCD dijet events and 100 000 signal events.

Events are required to have at least one R = 1.0 with pseudrapidity $|\eta| < 2.5$, and transverse momentum $p_\mathrm{T}^J > 1.2$ TeV. The top two leading $p_\mathrm{T}$ jets are selected and ordered by decreasing mass. In order to remove the turn on in the $m_{JJ}$ distribution arising from the jet selections, we only consider events with $m_{JJ} > 2.8$ TeV. To construct the training datasets we use varying amounts of signal events mixed in with the QCD dijet data.

To study the performance of our method in enhancing the sensitivity in a bump hunt, we use the input features proposed in Refs. [30–32, 35]. These are

$$m_{JJ}, m_{J_1}, \Delta m_J = m_{J_1} - m_{J_2}, \tau_{21}^{J_1}, \tau_{21}^{J_2}, \text{and } \Delta R_{JJ},$$

where $\tau_{21}$ is the N-subjettinness [45] ratio of a large radius jet, and $\Delta R_{JJ}$ is the angular separation of the two jets in the detector $\eta - \phi$ space.

## 3 Method

CURTAINsF4F follows the same motivation and approach as the original CURTAINs method presented in Ref. [35]. In bump hunt searches, data are categorised into non overlapping signal (SR) and side-band (SB) regions on a resonant distribution ($m_{JJ}$). In CURTAINs, a conditional Invertible Neural Network (cINN) is trained to learn the mapping from data drawn from one set of $m_{JJ}$ values to a target set of values. The cINN is trained using the SB regions and applied to transport data from the SB to the SR.

However, the approach improves upon CURTAINs by using a maximum likelihood loss on the transported data instead of an optimal transport loss between the batch of data and a batch of target data.

### 3.1 Flows for Flows architecture

A normalizing flow trained with maximum likelihood estimation requires an invertible neural network and a base distribution with an evaluable density. The standard choice for the base distribution is a standard normal distribution. The loss function for training the normalizing flow $f_\phi(z)$ from some distribution $x \sim X$ to the base distribution $z \sim p_{prior}$ is given from the change of variables formula

$$\log p_{\theta,\phi}(x) = \log p_\theta(f_\phi^{-1}(x)) - \log \left| \det(J_{f_\phi^{-1}(x)}) \right|,$$

where $J$ is the Jacobian of $f_\phi$. In the conditional case this extends to

$$\log p_{\theta,\phi}(x|c) = \log p_\theta \left( f_{\phi(c)}^{-1}(x|c) \right) - \log \left| \det(J_{f_{\phi(c)}^{-1}(x|c)}) \right|, \tag{1}$$

where $c$ are the conditional properties, $\theta$ are the parameters of the base distribution and $\phi$ are the parameter of the normalizing flow.

In Eq. (1), the base density term $\log p_\theta \left( f_{\phi(c)}^{-1}(x|c) \right)$ introduces a problem for training CURTAINs with maximum likelihood estimation. As the network is trained between two regions sampled from some non-analytically defined distribution, the probability of the transformed data is unknown. As a result, an optimal transport loss was used instead.

However, the base density of a normalizing flow can be chosen as any distribution for which the density can be evaluated. Therefore, we can train an additional normalizing flow to learn

the conditional density $p_\theta\left(f_{\phi(c)}^{-1}(x|c)\right)$ of the target data distribution. The normalising flow for learning the base distribution is trained in advance and is used to define the loss in Equation 1, and we refer to this as the base flow $f_\phi$. We can then train another normalising flow to learn the transformation between the two distiritbutions using what we refer to as the top flow $f_\gamma$. In CURTAINsF4F the conditional properties of the top flow are a function of the input ($x$) and target ($y$) conditional properties $c_x$ and $c_y$. For the base flow only a single conditional property $c$ is needed. The correspondence between the top normalizing flow and the base distributions in Flows for Flows is shown in Fig. 1.

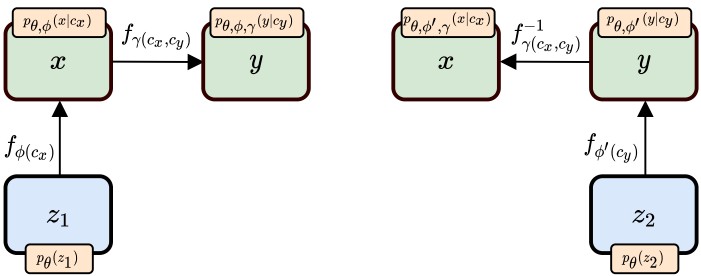

Figure 1: The Flows for Flows architecture for a conditional model [46]. Data $x$ ($y$) are drawn from the initial distribution with conditional values $c_x$ ($c_y$) and transformed to new values $c_y$ ($c_x$) in a cINN $f_\gamma(c_x, c_y)$ conditioned on $c_x$ and $c_y$. The probability of the transformed data points are evaluated using a second normalizing flow for the base distribution $f_{\phi'(c_y)}$ ($f_{\phi(c_x)}$). In the case where $x$ and $y$ are drawn from the same underlying distribution $p(x, c)$, the same base flow $f_\phi$ can be used. $\phi$ and $\phi'$ are the parameters of the base flow, and $\gamma$ represents the parameters of the top flow.

## 3.2 Training CURTAINsF4F

As with the original training method, CURTAINsF4F can be trained in both directions. The forward pass transforms data from low to higher target values of $m_{JJ}$, whereas the inverse pass transforms data from high to lower target values. When training between two distinct arbitrary distributions in both directions, a base flow is required for each distribution.

In principle, CURTAINsF4F could be trained between data drawn from the low $m_{JJ}$ SB (SB1) to the high $m_{JJ}$ SB (SB2), as performed with CURTAINs. However, as data no longer need to be compared to a target batch, it is possible to train with both SBs combined in a simplified training.

Data are drawn from both SBs and target $m_{JJ}$ values ($m_{target}$) are randomly assigned to each data point using all $m_{JJ}$ values in the batch. Data are passed through the network in a forward or inverse pass depending on whether $m_{target}$ is larger or smaller than their initial $m_{JJ}$ ($m_{input}$). The network is conditioned on a function of $m_{input}$ and $m_{target}$ with the two values ordered in ascending order ($f(m_{jj}^{low}, m_{jj}^{high})$). This function could be, for example, the concatenation of or difference between $m_{jj}^{low}$ and $m_{jj}^{high}$.

The probability term is evaluated using a single base flow trained on the data from SB1 and SB2. The loss for the batch is calculated from the average of the probabilities calculated from the forward and inverse passes. A schematic overview for a single pass through the network is shown in Fig. 2.

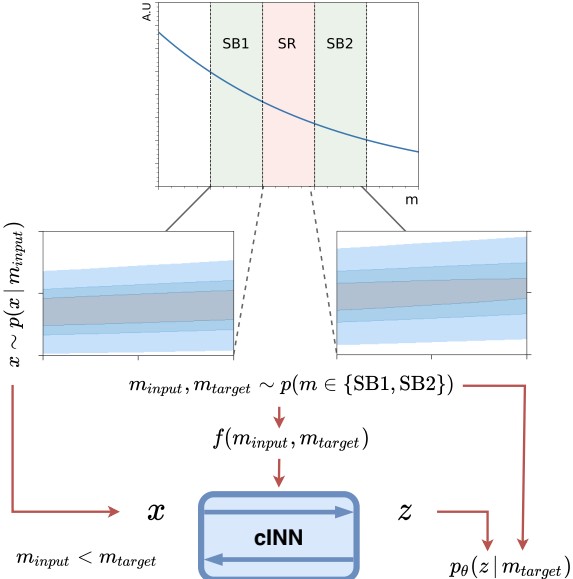

Figure 2: A schematic overview of the training procedure for CURTAINSF4F for an event where the target $m_{JJ}$ value is greater than the input value. A single conditional normalizing flow is used for the base flow, conditioned on the target $m_{JJ}$ value $m_{target}$, to determine $p_\theta(z|m_{target})$. The top normalizing flow is conditioned on a function of the input ($m_{input}$) and target ($m_{target}$) $m_{JJ}$ values. For the case where $m_{target} < m_{input}$, an inverse pass of the network is used and the conditioning property is calculated as $f(m_{target}, m_{input})$.

**Implementation**

The CURTAINSF4F architecture comprises two conditional normalizing flows, the base flow and the top flow. The base flow learns the conditional density of the training data which is used to train the top flow. The top flow in turn transforms data from initial values of $m_{JJ}$ to target values.

The base flow is trained on the side-band data with a standard normal distribution as the target prior. It is conditioned on $m_{JJ}$. The top flow is trained between data drawn from the side-bands. The transformation is conditioned on $\Delta m_{JJ} = m_{JJ}^{high} - m_{JJ}^{low}$. The base flow consists of ten autoregressive transformations using RQ splines, defined by four bins. The top flow consists of eight coupling transformations using RQ splines, defined by four bins. They are trained separately using the Adam optimiser and a cosine annealing learning rate scheduler. Each are trained for 100 epochs with a batch size of 256.

## 3.3 Generating background samples

To transform the data from the side-bands into the signal region, we assign target $m_{JJ}$ values corresponding to the SR to the data in each side-band. The target values $m_{target}$ are drawn from a function of the form

$$f(z) = p_1 (1-z)^{p_2} z^{p_3}, \tag{2}$$

where $z = m_{JJ}/\sqrt{s}$, with parameters $p_i$ learned by performing a fit to the side-band data. Data from SB1 (SB2) are transformed in a forwards (inverse) pass through the top flow with $\Delta m_{JJ}$.

The background template can be oversampled by assigning multiple target $m_{JJ}$ values to each data point. This has been found to improve the performance of CWOLA classifiers.

Due to the bidirectional nature of the cINNs, it is also possible to generate validation samples in regions further away from the SR. These outer-bands can be used to optimise the hyperparameters of the top flow in CURTAINsF4F.

## 3.4 Comparison to CURTAINs

CURTAINsF4F has a much simpler training procedure than in the original CURTAINs.

In CURTAINs, the Sinkhorn loss [47] was used to train the network, with the distance measured between a batch of data sampled from the target region and the transformed data. The target $m_{JJ}$ values for the transformed data were chosen to match the values in the target batch. However, there was no guarantee that the minimum distance corresponded to the pairing of the transformed event with the event in the target batch with the corresponding $m_{target}$ value. Furthermore, the loss itself ignored the $m_{JJ}$ values as the input data and target data in the batch with the corresponding $m_{JJ}$ target value are not necessarily events that should look the same for the same $m_{JJ}$ value. Although successful, this approach introduced a lot of stochasticity, and required a large number of epochs to converge.

Due to the new loss, the training in CURTAINsF4F no longer needs to be between two discrete regions. This has the added benefit that it removes the need for splitting the SBs and alternating between training CURTAINs between SB1 and SB2, and within each side-band.

Finally, in CURTAINs the network was trained and updated alternating batches in the forward and inverse directions. In CURTAINsF4F a single batch has both increasing and decreasing target $m_{JJ}$ values. As such the network weights are updated based on the average of the loss in both the forward and inverse directions for each individual batch.

Due to the additional base flow, CURTAINsF4F is no longer defined by a single model trained for each SR. This introduces an extra model which needs to be trained and optimised. We observe, however, that overall training both the base flow, and top flow between SB data is less than required to train the cINN in CURTAINs.

The additional time required to train the base flow can also be minimised. When training CURTAINsF4F for multiple SRs, a single base flow can be trained using all available data for all possible $m_{JJ}$ values. For each SR, the network would only be evaluated for values in SB1 and SB2, and only minimal bias would be introduced from data in the SR. This reduces the overall computational cost incurred when evaluating multiple signal regions.

## 3.5 Comparison to other approaches

This approach is one of several using normalizing flows as density models for background estimation for extending the sensitivity of bump hunts. Many of these methods, including CURTAINs and CURTAINsF4F, produce background samples for use with CWoLa bump hunting [30, 48]. In CWoLa bump hunting, classifiers are trained on data from a hypothesised signal enriched region (the SR) and a signal depleted region (the SBs). Cuts are applied onto the classifier score to reduce the amount of background and, in the presence of signal, enhance the sensitivity.

- In ANODE [31], conditional normalizing flows are trained to learn the probability of the signal and background from data drawn from the SBs and SR respectively. The normalizing flows are conditioned on $m_{JJ}$, and the ratio of the probabilities is used as a likelihood test.

- In CATHODE [32], a conditional normalizing flow is trained on all SB data conditioned on $m_{JJ}$. Samples with $m_{JJ}$ corresponding to the distribution of data in the SR (extrapolated from a side-band fit in $m_{JJ}$) are generated using the normalizing flow. These generated

samples form a synthetic background sample which together with the SR data are used in a CWoLa approach [30, 48].

- In FETA [37], the *Flows for Flows* approach is used to train a conditional normalizing flow between background data in a monte carlo (MC) simulated sample and the data in the side-bands, as a function of $m_{JJ}$. This flow is used to transport the MC events from the SR to the data space, and account for mismodelling observed in the simulated distributions. A CWoLa classifier is trained on the transported MC and the SR data.

- LaCATHODE [36] addresses the problem of distribution sculpting resulting from the choice of input features. Here the CWoLa classifier is trained on the base density of CATHODE, by first passing the SR data through the CATHODE conditional normalizing flow and using samples drawn from the prior base distribution. This approach is complementary to any method training a conditional normalizing flow, such as CURTAINsF4F and FETA.

- Although not applied in the context of bump hunt searches, ABCDNN [49] uses normalizing flows to extrapolate data from one region to another, similar to FETA. However, here the flows are trained with the maximum mean discrepancy loss, similar to the approach in CURTAINs though it does not interpolate to unknown conditional values.

## 4 Results

The main measure of performance for background estimation approaches is by how much they improve the sensitivity to a signal in a CWoLa bump hunt [30].

We define a SR centred on the signal process with a width of 400 GeV, which contains a substantially large fraction of the signal events. For CURTAINsF4F and CURTAINs, we use side-bands 200 GeV either side of the SR to train the methods. Only these regions are used to train the base flow for CURTAINsF4F. For CATHODE, the whole $m_{JJ}$ distribution either side of the SR is considered as the SB region. This corresponds to side-bands of widths 500 GeV and 900 GeV.

Weakly supervised classifiers are trained to separate the generated background samples from data in the SR. For CURTAINs, CURTAINsF4F, and CATHODE, an oversampling factor of four is used to generate the backround samples in the SR, at which point the performance reaches saturation. In CURTAINs and CURTAINsF4F this is with respect to the transported SB data, whereas for CATHODE it is based on the yields in the SR.

As a reference, a fully supervised classifier trained to separate the signal from background in this region, and an idealised classifier trained with a perfect background estimation are also shown. The idealised classifier is trained for both equal numbers of background in each class (Eq-Idealised) and an oversampled background (Over-Idealised).

A $k$-fold training strategy with five folds is employed to train all classifiers. Three fifths are used to train the classifier, with one fifth for validation and the final fifth as a hold out set. The classifiers comprise three hidden layers with 32 nodes and ReLU activations. They are trained for 20 epochs with the Adam optimiser and a batch size of 128. The initial learning rate is $10^{-4}$ but is annealed to zero following a cosine schedule.

### 4.1 Comparison of performance

Figure 3 shows the background rejection and significance improvement for CWoLa classifiers trained using the different background estimation models as the cut on the classifier is varied. Here 3,000 signal events have been added to the QCD dijet sample, of which 2,214 fall within the SR. CURTAINsF4F shows significant improvement over the original CURTAINs method, and

now matches the CATHODE performance across the majority of rejection and signal efficiency values. This is despite being trained on a much smaller range of data. CURTAINs still displays better significance improvement at very high rejection values, however this is in a region dominated by the statistical uncertainty.

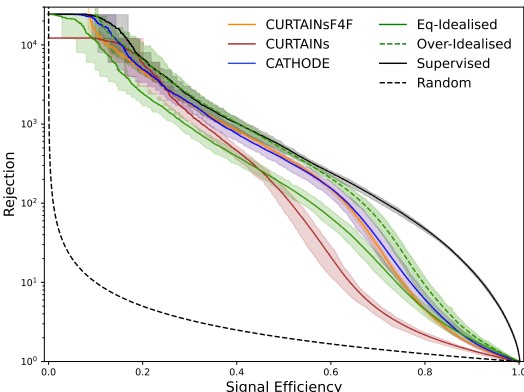
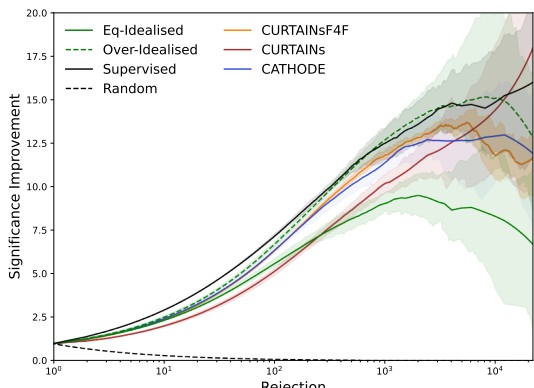

Figure 3: Background rejection as a function of signal efficiency (left) and significance improvement as a function of background rejection (right) for CURTAINs (red), CURTAINsF4F (orange), CATHODE (blue), Supervised (black), Eq-Idealised (green, solid), and Over-Idealised (green, dashed). All classifiers are trained on the sample with 3,000 injected signal events, using a signal region $3300 \leq m_{JJ} < 3700$ GeV. The lines show the mean value of fifty classifier trainings with different random seeds with the shaded band covering 68% uncertainty. A supervised classifier and two idealised classifiers are shown for reference.

In Fig. 4 the significance improvement for each method is calculated as a function of the number of signal events added to the sample. Here both the signal fraction and raw number of signal events in the SR are reported. The significance improvement is shown for two fixed background rejection values, rather than the maximum significance improvement, due to the sensitivity to fluctuations in the high background rejection regions where there are much lower statistics. The performance of CURTAINsF4F is improved across all levels of signal in comparison to the original CURTAINs method, and performs equally well as CATHODE.

## 4.2 Dependence on side-band width

In CURTAINsF4F, 200 GeV wide side-bands are used to train the networks and learn a local transformation. With leakage of signal into the side-bands or changing background composition, it could be beneficial to have narrower or wider SBs, and there is no set prescription for which is optimal. Figure 5 shows the impact on performance of varying the widths from 100 GeV up to all data not contained in the signal region (max width). For the 100 GeV wide side-bands a noticeable drop in performance is observed in the significance improvement and ROC curves. However at a background rejection of $\sim 10^3$ all other side-band widths have similar levels of rejection. At higher levels of background rejection training on larger side-bands, and thus more data, results in better performance than the default CURTAINsF4F model with widths of 200 GeV. It should be kept in mind that as the width of the side-bands increase, the required training time increases. For these comparisons no hyperparameter optimisation has been performed and the default values are used for all models.

In Fig. 6 the performance of CURTAINsF4F and CATHODE are shown for the case where each model is trained on either 200 GeV wide SBs or the max width. For CURTAINsF4F the difference in performance is mostly at high background rejection whereas CATHODE has a drop in performance at all values.

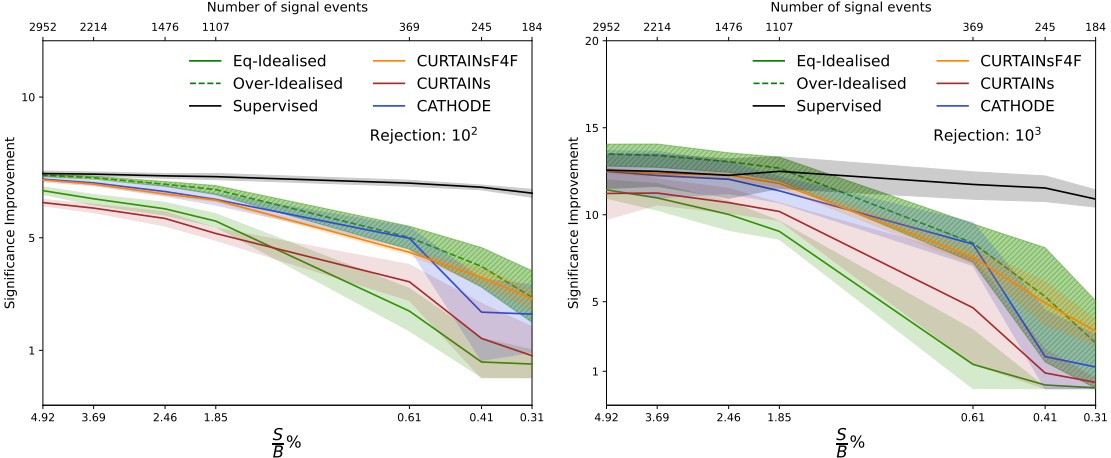

Figure 4: Significance improvement at a background rejection of $10^2$ (left) and $10^3$ (right) as a function of signal events in the signal region $3300 \leq m_{JJ} < 3700$ GeV, for CURTAINsF4F (orange), CURTAINs (red), CATHODE (blue), Supervised (black), Eq-Idealised (green, solid), and Over-Idealised (green, dashed). The lines show the mean value of fifty classifier trainings with different random seeds with the shaded band covering 68% uncertainty. A supervised classifier and two idealised classifiers are shown for reference.

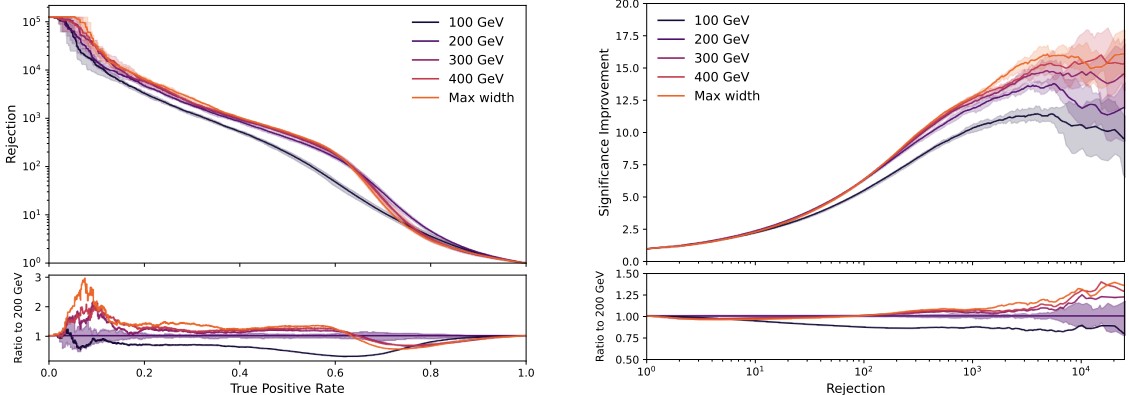

Figure 5: Background rejection as a function of signal efficiency (left) and significance improvement as a function of background rejection (right) for CURTAINsF4F trained with varying width side-bands, ranging from 100 GeV to the maximum width possible (SB1: 500 GeV, SB2: 900 GeV). All classifiers are trained on the sample with 3,000 injected signal events, using a signal region $3300 \leq m_{JJ} < 3700$ GeV. The lines show the mean value of fifty classifier trainings with different random seeds with the shaded band covering 68% uncertainty.

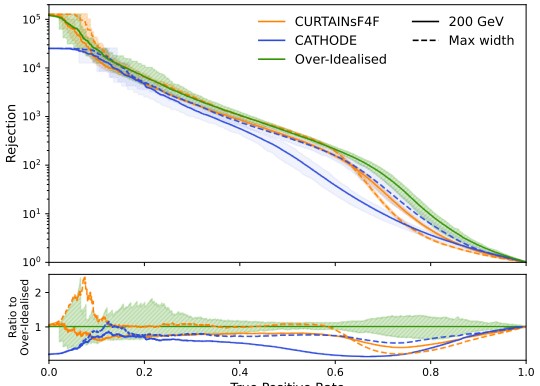
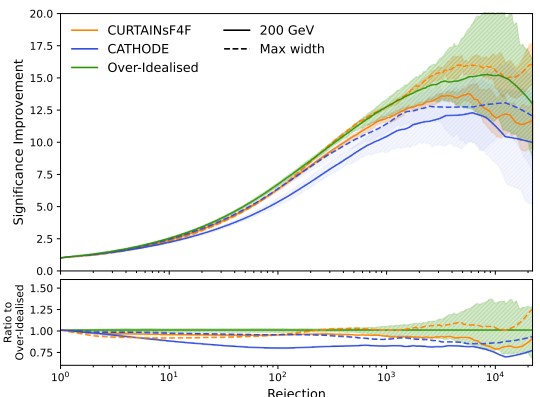

Figure 6: Background rejection as a function of signal efficiency (left) and significance improvement as a function of background rejection (right) for CURTAINsF4F (orange) and CATHODE (blue). Two side-band widths are used to train the two methods, 200 GeV side-bands (solid) and the maximum width (dashed, SB1: 500 GeV, SB2: 900 GeV). All classifiers are trained on the sample with 3,000 injected signal events, using a signal region $3300 \leq m_{JJ} < 3700$ GeV. The lines show the mean value of fifty classifier trainings with different random seeds with the shaded band covering 68% uncertainty. The Over-Idealised classifier (green) is shown for reference.

## 4.3 Required training time

For a bump hunt or sliding window search, numerous models need to be trained which can result in a high demand on computing resources. As a result, the granularity of a search may be restricted in line with overall computational time. Therefore, a key measure of methods like CURTAINsF4F and CATHODE is how quick the models are to train.

In Table 1 the required time to train the two approaches for one SR are shown for convergence and for one epoch. CATHODE has an advantage over CURTAINsF4F in that only one normalizing flow is trained. The total training time required for CURTAINsF4F is much reduced in comparison to CURTAINs and is slightly faster than CATHODE for the default configurations. This is largely due to CURTAINsF4F being able to generate a SR template by training on much narrower sidebands, whereas CATHODE trains on a much wider SB by default.

Table 1: Comparison of the required time to train CURTAINs, CURTAINsF4F, and CATHODE. All models are trained on the same hardware with epoch and total training time representative of using an NVIDIA® RTX 3080 graphics card. For CURTAINsF4F two numbers are shown for the epoch time and number of epochs due to the two normalizing flows which need to be trained. Default side-band widths are used for all models, around the nominal signal region.

|  | Time / epoch [s] | $N$ epochs | Total time [min] |
|---|---|---|---|
| CURTAINs | 10 | 1000 | 167 |
| CATHODE | 78 | 100 | 129 |
| CURTAINsF4F | 32/32 | 100/100 | 107 |

## 4.4 Reducing computational footprint

When applying the models to multiple signal regions in a bump hunt, new models need to be trained for each step. For CATHODE this involves training a complete model each time. However, due to the modular nature of CURTAINsF4F, if the base flow is trained on the whole spectrum, only the top flow needs to be trained with each step. It is important to note that the top flow only requires the densities of data in the sidebands (evaluated using the base flow) during the training, and any signal contamination in the data elsewhere is not an issue. With this modular training scheme, as soon as more than one SR is considered, CURTAINsF4F requires substantially less computational resources for a similar level of performance as can be seen in Table 2.

Additionally, the transformation learned by the top flow in CURTAINsF4F is known to be a smaller shift than for the base flow or in CATHODE. The top flow can thus also be optimised for speed without sacrificing as much performance and does not require the same expressive architecture as used by default.

The default CURTAINsF4F configuration is compared to an efficient implementation in which a single base flow is trained on all data, and the top flow is optimised for speed. The base flow has the same architecture as the default configuration. After a minimal hyperparameter scan, we find that the efficient top flow can be made with two coupling transformations using RQ splines, rather than eight, with each now defined by six bins instead of four. We find that the performance of the efficient configuration is comparable to the default configuration, with a significant reduction in training time. The top flow is trained for 20 epochs with a batch size of 256. All other hyperparameters remain unchanged and side-bands of 200 GeV are used to train the top flow and produce the background template in the SR. The potential reduction in computation time for using CURTAINsF4F in a sliding window search is presented in Table 2. With the efficient configuration more than one hundred signal regions can be evaluated with CURTAINsF4F transformers for the same computational cost as ten with the default configuration.

In Fig. 7 the significance improvement when using the efficient configuration is compared to the default CURTAINsF4F model for 3000 injected signal events. The performance as a function of the number of injected signal events is shown in Fig. 8. No significant decrease in performance is observed.

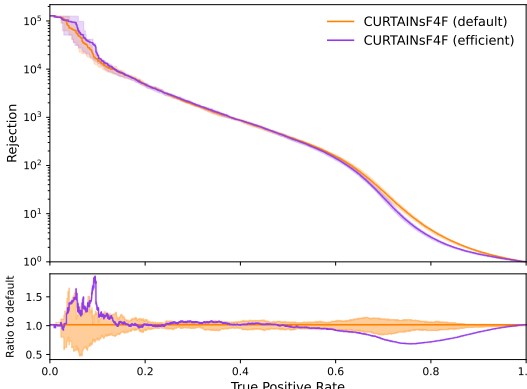 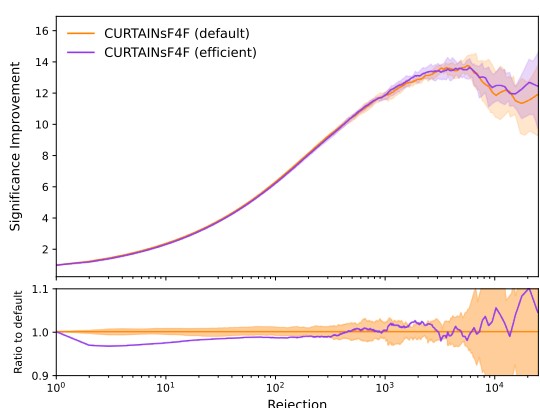

Figure 7: Background rejection as a function of signal efficiency (left) and significance improvement as a function of background rejection (right) for CURTAINsF4F using the default (orange) and efficient (purple) training configurations. All classifiers are trained on the sample with 3,000 injected signal events, using a signal region $3300 \leq m_{JJ} < 3700$ GeV. The lines show the mean value of fifty classifier trainings with different random seeds with the shaded band covering 68% uncertainty.

Table 2: Comparison of the required time to train the base flow and top flow in CURTAINsF4F. The default configuration comprises the base flow and top flow trained on 200 GeV side-bands. The efficient configuration has a single base flow trained on all data, and a top flow trained on 200 GeV side-bands and optimised for the fastest training time. All models are trained on the same hardware with epoch and total training time representative of using an NVIDIA® RTX 3080 graphics card. An extrapolation of the required total time to train a complete CURTAINsF4F model for one and ten signal regions are also shown for the two configurations. The extrapolated time for 125 signal regions is also shown for the efficient configuration, requiring less time than ten signal regions with the default configuration.

† Timing is for the nominal side-bands, this would vary as the signal region changes due to total number of training events.

| | Time / epoch [s] | $N$ epochs | Total time [min] |
|---|---|---|---|
| | Default | | |
| Base | 32.4† | 100 | 54 |
| Top flow | 31.5† | 100 | 53 |
| | One Signal Region | | 107 |
| (Extrapolated†) Ten Signal Region | | | 1070 |
| | Efficient | | |
| Base | 104.2 | 100 | 174 |
| Top flow | 21.3† | 20 | 7 |
| | One Signal Region | | 181 |
| (Extrapolated†) Ten Signal Region | | | 244 |
| (Extrapolated†) 125 Signal Region | | | 1049 |

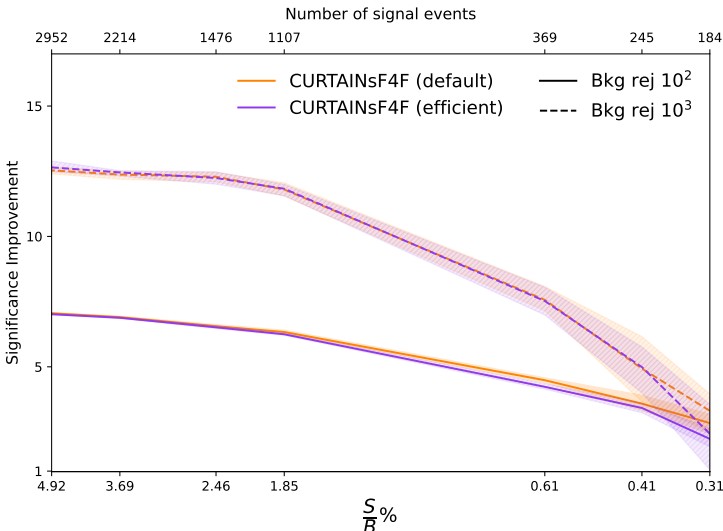

Figure 8: Significance improvement at a background rejection of $10^2$ and $10^3$ as a function of signal events in the signal region $3300 \leq m_{JJ} < 3700$ GeV for CURTAINsF4F using the default (orange) and efficient (purple) training configurations. The lines show the mean value of fifty classifier trainings with different random seeds with the shaded band covering 68% uncertainty.

# 5  Conclusions

In the original CURTAINS method, a distance based optimal transport loss was used to train a conditional invertible neural network. In this work we have shown that the performance can be improved significantly by moving to a maximum likelihood estimation loss, using the *Flows for Flows* methodology. The performance levels reached by CURTAINSF4F are state-of-the-art, and can do so training on less data from narrower side-bands than the previous state of the art.

By only modifying the training procedure, other advantages of CURTAINS are preserved. Additional validation regions further away from the signal region can be used to optimise the hyperparameters of both the normalizing flow and classification networks.

Furthermore, in order to address background sculpting resulting from the classifiers, the latent approach introduced in LACATHODE can be performed using the base flow. With the original CURTAINS method, an additional normalizing flow would need to be trained on the signal region data for each signal region.

Finally, for a single signal region CURTAINSF4F requires similar computing resources as other leading approaches, with almost half the required training time in comparison to the original CURTAINS method. However, when moving to a sliding window bump hunt, the overall computing resources required for CURTAINSF4F is reduced by a large factor. On the LHCO R&D dataset over one hundred signal regions can be trained for the same computing resources as otherwise required for ten signal regions. This could be of particular interest for large scale searches which are limited by the computational cost to cover a larger number of signal regions, such as those in Refs. [50, 51] amongst others.

## Acknowledgements

We would like to thank David Shih and Matt Buckley for valuable discussions at the ML4Jets 2022 conference at Rutgers, New Jersey, in particular on results of interest and potential applications. In addition, we would like to thank Radha Mastandrea, Ben Nachman and Kees Benkendorfer for useful discussions concerning the performance evaluation.

**Funding information**  The authors would like to acknowledge funding through the SNSF Sinergia grant called "Robust Deep Density Models for High-Energy Particle Physics and Solar Flare Analysis (RODEM)" with funding number CRSII5_193716 and the SNSF project grant 200020_212127 called "At the two upgrade frontiers: machine learning and the ITk Pixel detector.".

# A  Additional results

In Fig. 9 the maximum significance improvement for the default models is shown, rather than at fixed background rejection values.

An investigation on the sensitivity of CURTAINSF4F to the amount of oversampling is shown in Fig. 10. At a factor of four (default) the performance saturates.

In Table 3 the extrapolated times are computed using the faster top flow but with a new base flow for each signal region. Although there is a significant time improvement over the default configuration, the efficient implementation still almost three times faster for ten signal regions, and a factor of seven more signal regions can be trained in just over 1000 minutes.

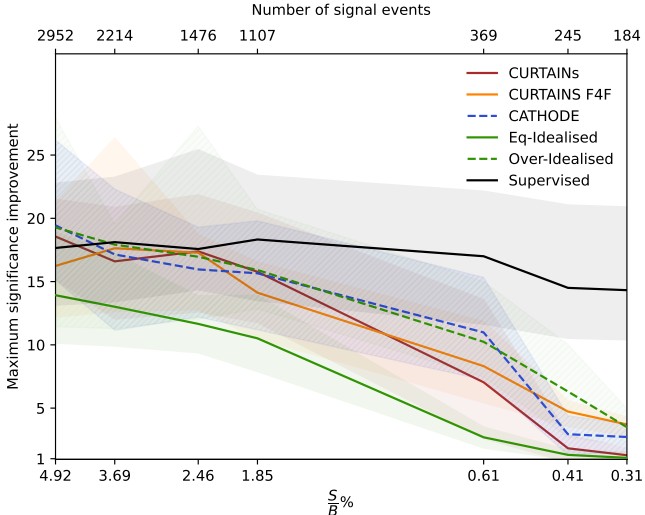

Figure 9: Maximum significance improvement as a function of signal events in the signal region $3300 \leq m_{JJ} < 3700$ GeV, for CURTAINs (red), CURTAINsF4F (orange), CATHODE (blue), Supervised (black), Eq-Idealised (green, solid), and Over-Idealised (green, dashed). The lines show the mean value of fifty classifier trainings with different random seeds with the shaded band covering 68% uncertainty. A supervised classifier and two idealised classifiers are shown for reference.

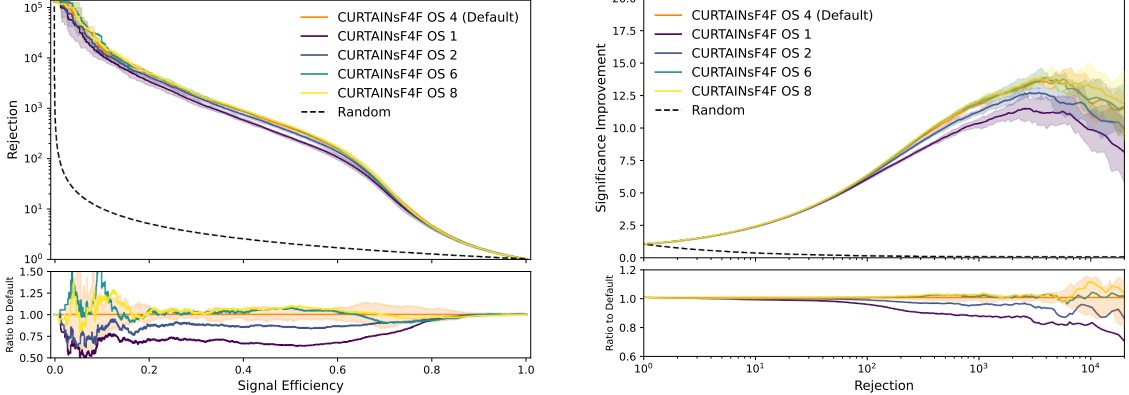

Figure 10: Background rejection as a function of signal efficiency (left) and significance improvement as a function of background rejection (right) for CURTAINsF4F trained with varying amounts of oversampling using 200 GeV side-bands All classifiers are trained on the sample with 3,000 injected signal events, using a signal region $3300 \leq m_{JJ} < 3700$ GeV. The lines show the mean value of fifty classifier trainings with different random seeds with the shaded band covering 68% uncertainty.

Table 3: The required time to train the base flow and top flow in CURTAINsF4F using the faster top flow but a base flow for each signal region. The base flow and top flow trained on 200 GeV side-bands. The models are trained on the same hardware with epoch and total training time representative of using an NVIDIA® RTX 3080 graphics card. An extrapolation of the required total time to train a complete CURTAINsF4F model for one and ten signal regions are also shown for the two configurations. †Timing is for the nominal side-bands, this would vary as the signal region changes due to total number of training events.

| | Time / epoch [s] | $N$ epochs | Total time [min] |
|---|---|---|---|
| | Faster | | |
| Base | 32.4† | 100 | 54 |
| Top flow | 21.3† | 20 | 7 |
| | One Signal Region | | 61 |
| (Extrapolated†) Ten Signal Region | | | 610 |
| (Extrapolated†) 17 Signal Region | | | 1037 |

## B    Hyperparameters

Table 4: Hyperparameters for training the flows in CURTAINsF4F.

| | Base distribution | Top flow (default) | Top flow (efficient) |
|---|---|---|---|
| Number of RQ splines | 10 | 8 | 2 |
| Number of bins per spline | 4 | 4 | 6 |
| Transformation | Autoregressive | Coupling | Coupling |
| Blocks per spline | 2 | 2 | 6 |
| Hidden nodes per block | 128 | 32 | 64 |
| Number of epochs | 100 | 100 | 20 |
| Batch size | 256 | 256 | 256 |
| Optimiser | Adam | Adam | Adam |
| Initial learning rate | 1e-4 | 1e-4 | 1e-4 |
| Cosine annealing | True | True | True |

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
