# Peer review of "CURTAINs Flows For Flows: Constructing Unobserved Regions with Maximum Likelihood Estimation"

_SciPost Physics, doi:SciPost Phys. 17, 046 (2024)_

## Round 1 · Referee Report · Ramon Winterhalder (Referee 1) · 2024-2-27

Strengths

  1. The CurtainsF4F approach is a significant improvement compared to the standard Curtains approach, both in terms of performance and elegance, due to its simplified loss and training.
  2. The paper nicely compares its method with other existing approaches to date (keeping in mind that the paper's submission date was May 2023!).
  3. It is nicely demonstrated that the method can be further improved regarding computational efficiency with very simple and little adjustments.

Weaknesses

  1. There are only minor points in terms of description and notation, which would improve the readability and make the paper even more accessible. Details are given in the requested changes.

Report

I very much appreciate the work and the effort being put into this manuscript. It is an excellent extension to previous works and is very much worth publishing. Before doing so, I would only like to ask for a minor revision as indicated in the requested changes.

Requested changes

  1. In the section around Eq. (1), it is unclear what the indices \theta and \phi indicate. Maybe this can be clarified with a sentence. Also, it is not clear to me why the Gaussian prior has some index \theta? Maybe I just do not fully understand the notation here.
  2. On the same line, in Figure 1, it should be clarified what the index \gamma stands for.
  3. In general, all Figures 3-8 would be easier to read if the plot labels were slightly larger. However, this is only a personal preference and should not be considered a critical point for not publishing.
  4. In section 4.3, while it has been mentioned before that Cathode has been trained on wider SB, it might be worth mentioning again that the difference in training size resulting from that is why Cathode has slightly longer training times compared to Curtains.
  5. In section 4.4, in contrast to the "default" setup, the base flow is trained on the entire spectrum, potentially including the relevant SR for later evaluation. When only reading this sentence, it is not directly obvious why this is ok. While the validity of this approach has become clear after rereading the approach and looking at the formulas, this might cause similar confusion to other readers. It would be worth clarifying this difference again and emphasizing why this is not a problem.
  6. Another point that is not directly clear to me is why your Top flow in the "efficiency" configuration with only 2 coupling blocks works in this scenario where it is acting on 6 or, more precisely, 5 + 1 (condition) dimensions. In general, as it has been shown in the first i-flow paper, to catch all correlations in a 5-6 dimensional space, you would require at least 6 (or 3 if your single coupling block modifies both partitions) coupling blocks, assuming optimal permutation between them. I assume you did some hyperparameter optimization on the number of coupling blocks? If so, it would be great to mention this. Also, the authors might comment on what this means if only 2 coupling blocks are sufficient to describe all relevant features.

  • validity: top
  • significance: top
  • originality: high
  • clarity: high
  • formatting: excellent
  • grammar: perfect

Author:  Debajyoti Sengupta  on 2024-07-05  [id 4603]

(in reply to Report 1 by Ramon Winterhalder on 2024-02-27)

Dear Ramon,

Thank you very much for your comments. Most of your comments were addressed during the resubmission. Apologies for not directly responding to this. Please find our responses below.

  1. In the section around Eq. (1), it is unclear what the indices \theta and \phi indicate. Maybe this can be clarified with a sentence. Also, it is not clear to me why the Gaussian prior has some index \theta? Maybe I just do not fully understand the notation here.

A brief text has been added after Eqn. 1 explaining what the different parameters refer to. \phi refer to the parameters of the flow, and \theta are the parameters of some generic base distribution (if N(0,1) this refers to the mean and std). A description of \gamma in Fig. 1 has also been added.

4. In section 4.3, while it has been mentioned before that Cathode has been trained on wider SB, it might be worth mentioning again that the difference in training size resulting from that is why Cathode has slightly longer training times compared to Curtains.

We agree and have added an explanation in this regard in sec. 4.3

5. In section 4.4, in contrast to the "default" setup, the base flow is trained on the entire spectrum, potentially including the relevant SR for later evaluation. When only reading this sentence, it is not directly obvious why this is ok. While the validity of this approach has become clear after rereading the approach and looking at the formulas, this might cause similar confusion to other readers. It would be worth clarifying this difference again and emphasizing why this is not a problem.

We have previously discussed this briefly at the end of section 3.4. We have since then added more text in section 4.4 justifying why this reasonable to do. Even though the base flow trains on the entire spectrum wherein the signal may potentially be localised somewhere, while training the top flow only the sideband data is used to query the base flow. As such, signal contamination elsewhere does not affect the performance.

6. Another point that is not directly clear to me is why your Top flow in the "efficiency" configuration with only 2 coupling blocks works in this scenario where it is acting on 6 or, more precisely, 5 + 1 (condition) dimensions. In general, as it has been shown in the first i-flow paper, to catch all correlations in a 5-6 dimensional space, you would require at least 6 (or 3 if your single coupling block modifies both partitions) coupling blocks, assuming optimal permutation between them. I assume you did some hyperparameter optimization on the number of coupling blocks? If so, it would be great to mention this. Also, the authors might comment on what this means if only 2 coupling blocks are sufficient to describe all relevant features.

The only optimisation done here was to see how lightweight the top flow can be made without significantly compromising on performance. The efficient top flow parameters were obtained as a result of that. This is now mentioned in section 4.4.

Debajyoti Sengupta, On behalf of the authors.

Author:  Debajyoti Sengupta  on 2024-07-05  [id 4602]

(in reply to Report 1 by Ramon Winterhalder on 2024-02-27)

Dear Ramon,

Thank you very much for your comments. Most of your comments were addressed during the resubmission. Apologies for not directly responding to this. Please find our responses below.

  1. In the section around Eq. (1), it is unclear what the indices \theta and \phi indicate. Maybe this can be clarified with a sentence. Also, it is not clear to me why the Gaussian prior has some index \theta? Maybe I just do not fully understand the notation here.

A brief text has been added after Eqn. 1 explaining what the different parameters refer to. \phi refer to the parameters of the flow, and \theta are the parameters of some generic base distribution (if N(0,1) this refers to the mean and std). A description of \gamma in Fig. 1 has also been added.

4. In section 4.3, while it has been mentioned before that Cathode has been trained on wider SB, it might be worth mentioning again that the difference in training size resulting from that is why Cathode has slightly longer training times compared to Curtains.

We agree and have added an explanation in this regard in sec. 4.3

5. In section 4.4, in contrast to the "default" setup, the base flow is trained on the entire spectrum, potentially including the relevant SR for later evaluation. When only reading this sentence, it is not directly obvious why this is ok. While the validity of this approach has become clear after rereading the approach and looking at the formulas, this might cause similar confusion to other readers. It would be worth clarifying this difference again and emphasizing why this is not a problem.

We have previously discussed this briefly at the end of section 3.4. We have since then added more text in section 4.4 justifying why this reasonable to do. Even though the base flow trains on the entire spectrum wherein the signal may potentially be localised somewhere, while training the top flow only the sideband data is used to query the base flow. As such, signal contamination elsewhere does not affect the performance.

6. Another point that is not directly clear to me is why your Top flow in the "efficiency" configuration with only 2 coupling blocks works in this scenario where it is acting on 6 or, more precisely, 5 + 1 (condition) dimensions. In general, as it has been shown in the first i-flow paper, to catch all correlations in a 5-6 dimensional space, you would require at least 6 (or 3 if your single coupling block modifies both partitions) coupling blocks, assuming optimal permutation between them. I assume you did some hyperparameter optimization on the number of coupling blocks? If so, it would be great to mention this. Also, the authors might comment on what this means if only 2 coupling blocks are sufficient to describe all relevant features.

The only optimisation done here was to see how lightweight the top flow can be made without significantly compromising on performance. The efficient top flow parameters were obtained as a result of that. This is now mentioned in section 4.4.

Debajyoti Sengupta, On behalf of the authors.

---

## Round 1 · Referee Report · Anonymous (Referee 2) · 2024-3-1

Report

I recommend the paper "CURTAINs Flows For Flows: Constructing Unobserved Regions with Maximum Likelihood Estimation" for publication in SciPost Physics after a (very) minor revision. The paper is a valuable addition to the literature on weakly supervised anomaly searches. CURTAINSF4F improves on the original CURTAINS idea and makes it competitive with other state of the art methods in terms of performance and has an edge in terms of run time as argued by the authors. Hence, CURTAINSF4F is becoming one of the standard methods for weakly supervised anomaly detection in bump hunt searches at the LHC and publication in SciPost Physics is clearly indicated. In the meanwhile, it has been well received by the community and has, for example, been used in a comparison of weakly supervised methods in 2307.11157. It is well written in general. The work and the corresponding results are convincingly presented.

Requested changes

A few points to be considered in a minor revision are listed in the following:

  1. The central reference for the "Flows for Flows" technique in the paper draft is 2211.02487. However, this paper seems to be no longer supported by the authors. It seems it has been replaced by the later publication 2309.06472. The situation should be clarified in the paper draft by referring to the relevant publication. The publication 2309.06472 contains several options for the generic "Flows for Flows" technique. The authors should make it clear which one has been used here.

  2. In section 2 you write "Events are required to have exactly two large radius jets". Maybe I am mistaken, but I do not think that there is a jet veto against a third hard jet in the data. I guess, just the two leading p_T jets are selected for further analysis and those are ordered in mass, correct?

  3. In the second paragraph of section 3.2 you write "Data are drawn from both SBs and target $m_{JJ}$ values (m_target) are randomly assigned to each data point using all $m_{JJ}$ values in the batch." Can you comment if this assignment is strictly necessary or if it would be also possible to draw target values from the $m_{JJ}$ distribution?

  4. In the third paragraph of section 3.2 you describe the averaging procedure for the loss and then refer to figure 2 for visualization. However, I do not think that the averaging is included in figure 2, is it? Otherwise I might have misunderstood the averaging procedure and some clarification might be helpful.

  5. At the end of section 3.4 you write "For each SR, the network would only be evaluated for values in SB1 and SB2, and no bias would be introduced from data in the SR." I perfectly believe that there is no noticeable effect in you analysis. Nevertheless, I find this statement too strong. I do not think that it is guaranteed that the fitting procedure of the conditional density estimator to the data in the side bands cannot be influenced by a potential signal in the signal region. After all the density is estimated as a whole, not specifically for a given $m_{JJ}$ value or region. If you agree you could make this statement a bit less strict.

  6. In the second paragraph of section 4 you write "We define a SR centred on the signal process with a width of 400 GeV, which contains nearly all of the signal events." Later on you state that it contains 2214 of 3000 events. So, "nearly all" is a slightly misleading statement.

  7. The name CURTAINSF4F appears in the abstract without being properly introduced. This could for example be easily done in the third line of the abstract: "We introduce CURTAINSF4F, a major... "

  8. You use "top flow" for the first time on the bottom of page 3 as if this is a well-known phrase. I think this phrase should be properly introduced. On page 4 you also call it "transformer flow".

  9. There are a few typos: better significant improvement -> better significance improvement (section 4.1) performs equally as well as CATHODE -> performs equally well as CATHODE (section 4.1) the require time -> the required time (section 4.3)

  • validity: -
  • significance: -
  • originality: -
  • clarity: -
  • formatting: -
  • grammar: -

Author:  Debajyoti Sengupta  on 2024-05-02  [id 4464]

(in reply to Report 2 on 2024-03-01)
Category:
answer to question
correction
pointer to related literature

We thank the reviewer for their comments on the manuscript. Please find our responses to the comments inline.

  1. The central reference for the "Flows for Flows" technique in the paper draft is 2211.02487. However, this paper seems to be no longer supported by the authors. It seems it has been replaced by the later publication 2309.06472. The situation should be clarified in the paper draft by referring to the relevant publication. The publication 2309.06472 contains several options for the generic "Flows for Flows" technique. The authors should make it clear which one has been used here.

We have updated the ref. to reflect the newest version of the work. The text now says this work is built on Flows for Flows which is uniquely identified in the Table 1 of the ref.. Thus there should hopefully be no ambiguity as to which option was used.

  1. In section 2 you write "Events are required to have exactly two large radius jets". Maybe I am mistaken, but I do not think that there is a jet veto against a third hard jet in the data. I guess, just the two leading p_T jets are selected for further analysis and those are ordered in mass, correct?

Thanks for the correction, this was indeed an oversight on our end. The text has been edited to reflect the correct event selection.

  1. In the second paragraph of section 3.2 you write "Data are drawn from both SBs and target mJJ values (m_target) are randomly assigned to each data point using all mJJ values in the batch." Can you comment if this assignment is strictly necessary or if it would be also possible to draw target values from the mJJ distribution?

The network is trained to learn a conditional map p(x|m) for all the features, and as such it is necessary to have the target features be associated to theit mJJ value. A random sampling of mJJ would break this correlation and the learnt map may not be useful. During inference, because the top flow has learnt the conditional map - it can then be used to generate the template for any pair of initial and target mjj values.

  1. In the third paragraph of section 3.2 you describe the averaging procedure for the loss and then refer to figure 2 for visualization. However, I do not think that the averaging is included in figure 2, is it? Otherwise I might have misunderstood the averaging procedure and some clarification might be helpful.

The text has now been updated to say that the figure only shows one pass through the network.

  1. At the end of section 3.4 you write "For each SR, the network would only be evaluated for values in SB1 and SB2, and no bias would be introduced from data in the SR." I perfectly believe that there is no noticeable effect in you analysis. Nevertheless, I find this statement too strong. I do not think that it is guaranteed that the fitting procedure of the conditional density estimator to the data in the side bands cannot be influenced by a potential signal in the signal region. After all the density is estimated as a whole, not specifically for a given mJJ value or region. If you agree you could make this statement a bit less strict.

We have updated the text to make the statement a little less bold than before.

  1. In the second paragraph of section 4 you write "We define a SR centred on the signal process with a width of 400 GeV, which contains nearly all of the signal events." Later on you state that it contains 2214 of 3000 events. So, "nearly all" is a slightly misleading statement.

Updated the text to say "a substantial fraction of"

  1. The name CURTAINSF4F appears in the abstract without being properly introduced. This could for example be easily done in the third line of the abstract: "We introduce CURTAINSF4F, a major... "

Updated the text in abstract.

  1. You use "top flow" for the first time on the bottom of page 3 as if this is a well-known phrase. I think this phrase should be properly introduced. On page 4 you also call it "transformer flow".

All references to Transformer flow have been rectified. Furthermore a small introduction to base flow and top flow has been added.

  1. There are a few typos: better significant improvement -> better significance improvement (section 4.1) performs equally as well as CATHODE -> performs equally well as CATHODE (section 4.1) the require time -> the required time (section 4.3)

Rectified.

Attachment:

main_sp_compressed.pdf

---

## Editorial Decision

published